# Protection and Connection: Negating Depression and Suicidality among Bullied, LGBTQ Youth

**DOI:** 10.3390/ijerph20146388

**Published:** 2023-07-18

**Authors:** Lindsay Kahle Semprevivo

**Affiliations:** Department of Criminal Justice, Radford University, Radford, VA 24142, USA; lsempreivov@radford.edu

**Keywords:** depression, suicidality, bullying, LGBTQ youth

## Abstract

Lesbian, gay, bisexual, transgender, queer and/or questioning (LGBTQ) youth are particularly at risk of bullying and other forms of violence, and the myriad of risk factors associated with instances of victimization. Interdisciplinary research finds that certain protective factors—biological, psychological, familial, or community-level characteristics that reduce the impact of risk and problematic outcomes—mitigate the effects of victimization. Using data from the 2019 Nashville Youth Risk Behavioral Surveillance System (YRBSS), this study examines the effects of bullying and electronic bullying on LGBTQ (*n* = 303) and heterosexual/cisgender (*n* = 1104) 9th to 12th-grade students’ depression and suicidality, and the role that protective factors play in mitigating these effects. Logistic regression results show that students who feel safe at school, feel valued by their community, and seek help are less likely to report depression and suicidality overall, when they are LGBTQ, and when they are bullied. These findings point to the importance of solidifying personal, school, and community-level support systems for youth, especially LGBTQ youth.

## 1. Introduction

Lesbian, gay, bisexual, transgender, queer and/or questioning (LGBTQ) youth are a growing population in the United States. The estimated number of LGBQ youth aged 13–17 averages around 9.5%, and the number of transgender youth averages around 0.7% [1]. Recent news coverage highlights that more young people identify as LGBTQ than ever before, as high as one in six among the Gen Z population [2,3]. As a result, society must continue moving towards greater visibility and acceptance, providing better networks of support, and addressing the issues that often plague LGBTQ young people.

Youth who identify as LGBTQ face a myriad of risk factors which are, in turn, associated with a variety of negative health outcomes [4]. Stigma and discrimination manifest in multiple ways, across different facets of LGBTQ lives [5,6]. Institutional discrimination in the most foundational aspects of children’s lives, such as school systems, places of worship, peer groups, the law, and even the family, create uniquely difficult circumstances for LGBTQ youth that often result in (not just physical) violence and victimization [7]. Lesbian, gay, bisexual, transgender, queer, and/or questioning youth have some of the highest reported rates of victimization, including various types of harassment and bullying, both in person and online, as well as physical assault and sexual violence [8,9]. According to the 2021 National School Climate Survey, 76% of LGBTQ students experienced verbal harassment, 31% experienced physical harassment, 13% were physically assaulted, 54% were sexually harassed, and between 30 and 37% of students were harassed online based on their sexual orientation, gender, or gender expression [8]. Hostile climates and behaviors such as these often produce negative physical, mental, and/or physiological harm among young people experiencing them. Victimization (and bullying in particular) is one of the highest risk factors for poor grades and educational outcomes, truancy, substance use, psychological distress, and even suicide [10,11,12,13].

A clear risk for poor mental health exists among LGBTQ students because of the many risk factors they face, especially victimization. According to the major trends in the 2015–2019 Youth Risk Behavior Survey data, 60 to 66% of LGB students reported feeling sad or hopeless, compared to 26–32% of heterosexual youth. Lesbian, gay, and bisexual students also had greater chances of being at risk of suicide than heterosexual students, and these also varied by sex. Over half of transgender students (53.1%) reported feeling sad or hopeless almost every day for two weeks or more, which was well above cisgender male and female students. In addition, the report also found that planned suicide attempts were higher among transgender students than cisgender male and female students, with 39.3% of transgender students reporting that they had made a suicide plan within the past year [14]. There is a clear need for multilevel mechanisms of support for LGBTQ youth.

The National School Climate Survey and other evidence-based research note the need for supportive environments, because they help reduce victimization and equip LGBTQ youth with the proper tools to build resilience and overcome a possible trajectory towards the negative outcomes that may occur as a result of victimization. Protective factors are different characteristics associated with the possible mitigation, or countering, of negative outcomes and risk factors [15]. Protective factors exist at the individual, community, and familial levels, and are seen as protections against the negative stimuli that young people may encounter. They allow them to succeed ‘despite’ the presence of risk [15].

The protective factors literature makes an important point of turning the conversation away from focusing solely on risk factors, pushing instead to focus on sources of support and building positive traits and resiliency. Protective factors are discussed in a variety of interdisciplinary contexts and ways, including classroom environments and educational outcomes, child welfare and wellbeing, the prevention of abuse and neglect, and positive life-course adjustment and coping [16,17,18,19]. With regard to adolescent mental health and wellbeing in particular, research notes that self-esteem, strong family and peer relationships, social support, and even finding meaning in life help mitigate particular risk factors and have a significantly positive influence on adolescent mental health [20,21]. Overall, there is noticeable overlap in protective factors and the risk that they help to improve and mitigate, but that does not negate their importance.

A meta-analysis on bullying and cyberbullying found that different types of individual, peer, familial, school, and community factors all acted as significant protective factors against bullying. Factors such as self-oriented personal competencies (i.e., self-esteem, cognition, emotional management, life satisfaction, openness, and consciousness), peer status (i.e., influence and support), a positive school climate, and positive parenting and supervision all contributed to lower levels of bullying and victimization [22]. Additional research found that even aspects like family context were negatively associated with bullying behaviors and victimization [23].

One of the biggest components of the LGBTQ protective factors literature deals with promoting resilience, or the capability that people have to overcome adversity. According to The Trevor Project, protective factors like social support, role models, and inclusive policies all help mitigate the stigma and isolation, bullying, and overt discrimination that can make LGBTQ youth vulnerable to a range of poor mental health outcomes [24]. A meta-analysis on LGBTI+ wellbeing found that things such as interpersonal relationships with parents, peers, and providers, as well as community relations (online, faith, and cultural) and GSA’s, all served as protective factors for queer youth [24].

To date, few studies have investigated protective factors using the YRBSS data. The purpose of this study is to explore whether different individual, school, and community protective factors moderate depression and suicidality as a result of bullying, particularly among LGBTQ youth. More specifically, this study utilizes the 2019 YRBSS survey data in order to answer the following research questions. First, do these protective factors moderate depression and suicidality among youth in general? Second, do these protective factors moderate depression and suicidality among LGBTQ youth in particular? Third, do these protective factors moderate depression and suicidality among youth who are bullied at school? Finally, do these protective factors moderate depression and suicidality among bullied, LGBTQ youth?

## 2. Materials and Methods

In order to investigate the role that these factors may play in moderating youth depression and suicidality, this project utilizes data from the 2019 Youth Risk Behavior Surveillance System (YRBSS). The YRBSS is a survey developed by the Centers for Disease Control and Prevention (CDC) in order to monitor health risk behaviors that contribute to causes of death, disability, and social problems among youth in the United States. The sampling frame included all public, parochial, and other nonpublic schools across all 50 states and the District of Columbia, and used a three-stage cluster sampling design to produce a nationally representative sample of students across 136 schools [25]. In 2015, the YRBSS added two new questions to the questionnaire concerning sexual identity and sexual behavior, and in 2017, 19 (state and district level) sites were permitted to pilot questions regarding transgender identity [26]. In addition, the questions used to measure protective factors are considered “optional” questionnaire items, thus are included in only a handful of sites across the United States. State- and district-level data that did not include the protective factor questions, or students’ sexual orientation or gender identity, were excluded. This resulted in a representative district-level sample from Nashville, Tennessee. The data did not require methods for dealing with missing cases. The final sample size was 1407 (303 LGBTQ and 1104 heterosexual) 9th to 12th-grade students.

### 2.1. Measures

#### 2.1.1. Independent Variables

**LGBTQ.** Sexual orientation was measured through self-identification, in which respondents were asked, “which of the following best describes you?” Response items ranged from (A) heterosexual (straight), (B) gay or lesbian, (C) bisexual, and (D) not sure (i.e., questioning). Similar to other studies [6], questioning students reported an equally high risk of negative outcomes compared to LGB students, thus were included in the measure. Data regarding transgender students were operationalized using the following question: “some people describe themselves as transgender when their sex at birth does not match the way they think or feel about their gender. Are you transgender?” Response items ranged from (A) no, I am not transgender, (B) yes, I am transgender, (C) I am not sure if I am transgender, and (D) I do not know what this question is asking. Responses to this item were then dichotomized as (0) not transgender/not sure/do not know what question is asking, and (1) transgender, in order to indicate whether a student was transgender. A composite variable, *LGBTQ,* was then created that included students who identified as (1) gay or lesbian, bisexual, not sure (i.e., questioning) and/or transgender, or (0) heterosexual and/or not transgender/not sure/do not know what question is asking.

**Victimization.** The independent variable *bullying* was operationalized using the question, “during the past 12 months, have you ever been bullied on school property?” and responses were (1) yes, and (2) no. This variable was then re-coded into (0) no, and (1) yes, in order to facilitate the logistic regression models.

**Protective factors.** Several questions measured protective factors at the school, community, and individual levels. Two variables measured protective factors at the school level. *Feeling safe at school* was operationalized using the question, “how often do you feel safe and secure at school?” Responses ranged from (A) never, (B) rarely, (C) sometimes, (D) most of the time, and (E) always, and were then dichotomized by the percentage of students who most of the time or always feel safe and secure at school as (0) no, and (1) yes. *Feels close to people at school* was operationalized using the question, “do you agree or disagree that you feel close to people at you school?” Responses ranged from (A) strongly agree, (B) agree, (C) not sure, (D) disagree, and (E) strongly disagree. They were then dichotomized by the percentage of students who strongly agree or agree that they feel close to people at their school as (0) no, and (1) yes. One variable measured protective factors at the community level. *Matter to their community* was operationalized using the following question: “do you agree or disagree that in your community you feel like you matter to people?” Responses ranged from (A) strongly agree, (B) agree, (C) not sure, (D) disagree, and (E) strongly disagree. They were then dichotomized by the percentage of students who strongly agree or agree that in their community they feel like they matter to people as (0) no, and (1) yes. Two variables measured help-seeking behaviors among individuals. The variable *seeks help when depressed* was operationalized using the following question: “when you feel sad, empty, hopeless, angry, or anxious, how often do you get the kind of help you need?” Responses ranged from (A) I do not feel sad, empty, hopeless, angry, or anxious, (B) never, (C) rarely, (D) sometimes, (E) most of the time, and (F) always. They were then dichotomized by the percentage of students who report having felt sad, empty, hopeless, angry, or anxious as (0) no, and (1) yes. Finally, *seeks help from adults* was operationalized using the following question: “besides your parents, how many adults would you feel comfortable seeking help from if you had an important question affecting your life?” Responses ranged from (A) 0 adults, (B) 1 adult, (C) 2 adults, (D) 3 adults, (E) 4 adults, and (F) 5 or more adults. They were then dichotomized by the percentage of students who would feel comfortable seeking help from one or more adults besides their parents if they had an important question affecting their life as (0) no, and (1) yes.

#### 2.1.2. Dependent Variables

This study measures the outcomes of depression and suicidality. *Depression* was operationalized using the following question: “during the past 12 months, did you ever feel so sad or hopeless almost every day for two weeks or more in a row that you stopped doing some usual activities?” Responses ranged from (0) no, or (1) yes. *Suicidality* was created from the following questions: “during the past 12 months, did you ever seriously consider attempting suicide, “during the past 12 months, did you make a plan about how you would attempt suicide,” and “during the past 12 months, how many times did you actually attempt suicide?” While the first two question responses ranged from (0) no, or (1) yes, the second question ranged from (A) 0 times, (B) 1 time, (C) 2 or 3 times, (D) 4 or 5 times, or (E) 6 or more times. The second question was dichotomized as (0) no, or (1) yes. A composite variable (α= 0.810) was created for a total suicidality measure.

#### 2.1.3. Controls

*Female* was operationalized using the following question: “what is your sex?” Response items were dichotomized as (0) male and (1) female. *Grade* was operationalized using the question, “In what grade are you?” Responses ranged from (A) 9th, (B) 10th, (C) 11th, (D) 12th, and (E) ungraded or other grade. Race and ethnicity were operationalized using the following question: “how do you describe yourself?”, and responses were coded into five dichotomous variables, with white youth excluded as the comparison group: (1) *American Indian/Alaska Native/Native Hawaiian/other Pacific Islander* and (0) not American Indian/Alaska Native/Native Hawaiian/other Pacific Islander; (1) *Asian* and (0) not Asian; (1) *Black or African American* and (0) not Black or African American; (1) *Hispanic/Latino* and (0) not Hispanic/Latino; and (1) *Multiple races (non-Hispanic)* and (0) not Multiple races (non-Hispanic).

#### 2.1.4. Data Analysis

Using binary logistic regression, this study explored the role of protective factors in how they moderate depression and suicidality generally, among LGBTQ youth specifically, among bullied youth, and among bullied LGBTQ youth. Thus, the analyses proceed in several steps. Table 1 presents the descriptive statistics for the variables in the study. Table 2 presents the odds ratios for protective factors, depression (Model 1), and suicidality (Model 2) generally across the sample of students. Table 3 presents the odds ratios for protective factors among LGBTQ youth, specifically; Models 3 and 4 regress LGBTQ identity, controls and the protective factors on depression, and Models 5 and 6 regress LGBTQ identity, controls and the protective factors on suicidality. Similarly, Table 4 presents the odds ratios for protective factors among bullied youth, specifically. Models 7 and 9 regress bullying on depression and suicidality, and Models 8 and 10 regress bullying, protective factors, and controls on depression and suicidality. Finally, Table 5 and Table 6 present the odds ratios for protective factors, depression, and suicidality among bullied LGBTQ youth. Model 11 regresses bullying, LGBTQ identity, and controls on depression; Model 12 regresses LGBTQ bullying, LGBTQ identity, and controls on depression; and Model 13 (which is considered the full model), regresses bullying, LGBTQ bullying, LGBTQ identity, protective factors, and controls on depression. Models 14, 15, and 16 repeat these regressions for the dependent variable of suicidality.

## 3. Results

### 3.1. Descriptive Statistics

Table 1 presents the descriptive statistics for the variables in the study. Overall, about 22% of the sample identified as lesbian, gay, bisexual, questioning, and/or transgender, and 18% reported being bullied at school within the past 12 months. In terms of school, community, and individual protective factors, 60% of students reported feeling safe at school, 55% of students felt close to people at their school, 47% felt that they mattered to their community, 26% of students most of the time or always sought help when they felt sad, empty, hopeless, angry, or anxious, while 68% felt comfortable seeking help from one or more adults if they had an important question affecting their life.

Among students in the sample, 38% reported feeling sad or hopeless almost every day for two or more weeks in a row to the point that they stopped performing some usual activities, and 28% of students reported considering, planning, or attempting (at least once) suicide within the 12 months prior to the survey. With regard to controls, 48% of the sample identified as female, the mean grade level of the students was 10th grade, 1% of the sample was American Indian or Alaska Native, 5% of the sample was Asian, 36% of the sample was Black or African American, less than 1% were Native Hawaiian or other Pacific Islander, 12% were Hispanic or Latino, and 6% identified as Multi-race Non-Hispanic.

**Table 1 ijerph-20-06388-t001:** Descriptive statistics.

**Independent Variables:**	**Range**	**Frequency**	**X (bar)**	**SD**
LGBTQ	0–1	303	0.22	0.411
**Victimization**	**Range**	**Frequency**	**X (bar)**	**SD**
Bullying	0–1	251	0.18	0.385
**Protective Factors**	**Range**	**Frequency**	**X (bar)**	**SD**
Feels safe at school	0–1	839	0.60	0.490
Feels close to people at school	0–1	780	0.55	0.495
Matter to their community	0–1	658	0.47	0.500
Seeks help when depressed	0–1	371	0.26	0.396
Seeks help from Adults	0–1	961	0.68	0.456
**Dependent Variables:**	**Range**	**Frequency**	**X (bar)**	**SD**
Depression	0–1	538	0.38	0.488
Suicidality	0–1	389	0.28	0.467
**Controls:**	**Range**	**Frequency**	**X (bar)**	**SD**
Female	0–1	681	0.48	0.500
Grade	9th–12th	1389	10.2	1.094
American Indian/Alaska Native	0–1	12	0.01	0.094
Asian	0–1	68	0.05	0.218
Black or African American	0–1	485	0.36	0.479
Native Hawaiian/Other PI	0–1	6	0.004	0.066
Hispanic/Latino	0–1	168	0.12	0.329
Multi-race non-Hispanic	0–1	85	0.06	0.242

### 3.2. Protective Factors, Depression, and Suicidality

Table 2 shows the effects of protective factors on students’ depression and suicidality in general. Three protective factors significantly reduced depression and four protective factors reduced suicidality among youth in this study. Students who feel safe at school (*b* = −0.521, *OR* = 0.594, *p* ≤ 0.001), feel that they matter to their community (*b* = −0.718, *OR* = 0.488, *p* ≤ 0.001), and those who seek help most of the time or always when they feel sad, empty, hopeless, angry, or anxious (*b* = −0.603, *OR* = 0.547, *p* ≤ 0.001) have lower odds of feeling sad or hopeless than those who do not. In turn, students who feel safe at school (*b* = −0.841, *OR* = 0.431, *p* ≤ 0.001), feel that they matter to their community (*b* = −348, *OR* = 0.706, *p* ≤ 0.05), those who seek help most of the time or always when they feel sad, empty, hopeless, angry, or anxious (*b* = −0.517, *OR* = 0.596, *p* ≤ 0.001) and those who feel comfortable seeking help from one or more adults (*b* = −0.325, *OR* = 0.723, *p* ≤ 0.05) have lower odds of considering, planning, or attempting suicide than students who do not.

With regard to the controls, certain facets of identity present risk and protection against reporting depression and suicidality. Overall, females (*b* = 0.680, *OR* = 1.974, *p* ≤ 0.001) have higher odds of reporting depression and higher odds (*b* = 0.355, *OR* = 1.426, *p* ≤ 0.05) of reporting suicidality than males. Overall, Black or African American (*b* = −0.340, *OR* = 0.712, *p* ≤ 0.05) students are less likely to report depression and those who identify as Asian (*b* = −0.757, *OR* = 0.469, *p* ≤ 0.05) are less likely to report suicidality than white students. In turn, those who identify as Multi-race Non-Hispanic (*b* = 0.528, *OR* = 1.696, *p* ≤ 0.05) have higher odds of reporting depression, and those who identify as Native Hawaiian or Other Pacific Islander (*b* = 2.289, *OR* = 9.861, *p* ≤ 0.001) have significantly higher odds of reporting suicidality than white students. Similar trends persist throughout the models, where gender and race and ethnicity present certain risk and protective factors against depression and suicidality.

**Table 2 ijerph-20-06388-t002:** Protective Factors, depression, and suicidality.

	Depression			Suicidality		
	Model 1			Model 2		
	*b*	(*SE*)	*OR*	*b*	(*SE*)	*OR*
Safe at school	−0.521	0.126	0.594 ***	−0.841	0.139	0.431 ***
Close at school	−0.180	0.136	0.835	−0.245	0.151	0.783
Community	−0.718	0.138	0.488 ***	−0.348	0.154	0.706 *
Depressed help	−0.603	0.15	0.547 ***	−0.517	0.169	0.596 ***
Adult Help	−0.198	0.136	0.820	−0.325	0.149	0.723 *
Female	0.680	0.125	1.974 ***	0.355	0.139	1.426 *
Grade	0.023	0.057	1.023	−0.063	0.063	0.939
AI/AN	−0.879	0.841	0.415	0.586	0.796	1.797
Asian	−0.441	0.294	0.644	−0.757	0.375	0.469 *
Black/A-A	−0.340	0.143	0.712 *	0.102	0.156	1.107
NH/Other PI	−0.13	0.945	0.878	2.289	1.165	9.861 *
Hispanic/Latino	−0.381	0.199	0.683	−0.372	0.235	0.69
MR non-Hispanic	0.528	0.257	1.696 *	0.353	0.275	1.424
Chi-square	158.686			110.788		
Log Likelihood	1546.065			1275.12		
Nagelkerke R^2^	0.159			0.133		
Constant	0.336		1.400	0.274		1.315

*** *p* ≤ 0.001; * *p* ≤ 0.05.

### 3.3. LGBTQ Youth

Table 3 presents the effects of LGBTQ identity on depression and suicidality, as well as the effects of protective factors on these outcomes. Models 3 and 5 show that youth who identify as LGBTQ have higher odds of reporting depression (*b* = 1.073, *OR* = 2.925, *p* ≤ 0.001) and suicidality (*b* = 1.317, *OR* = 3.734, *p* ≤ 0.001) than those who do not identify as LGBTQ. Models 4 and 6 show that certain protective factors also matter with regard to LGBTQ identity, depression and suicidality. Students who feel safe at school (*b* = −0.483, *OR* = 0.617, *p* ≤ 0.001), those who feel like they matter to their community (*b* = −0.694, *OR* = 0.500, *p* ≤ 0.001), and those who seek help most of the time or always when they feel sad, empty, hopeless, angry, or anxious (*b* = −0.625, *OR* = 0.535, *p* ≤ 0.001) have lower odds of reporting depression than students who do not. Similarly, students who feel safe at school (*b* = −0.830, *OR* = 0.436, *p* ≤ 0.001), those who seek help most of the time or always when they feel sad, empty, hopeless, angry, or anxious (*b* = −0.527, *OR* = 0.591, *p* ≤ 0.001), and those who feel comfortable seeking help from one or more adults (*b* = −0.323, *OR* = 0.724, *p* ≤ 0.05) have lower odds of reporting suicidality than students who do not. These trends are similar to the patterns presented in the overall depression and suicidality models presented in Table 2. When considering all of these factors, it appears that protective factors minimally moderate the effects of LGBTQ identity on depression and suicidality by slightly reducing the odds ratios of each.

**Table 3 ijerph-20-06388-t003:** Protective factors, depression, suicidality, and LGBTQ youth.

	Depression						Suicidality					
	Model 3			Model 4			Model 5			Model 6		
	*b*	(*SE*)	*OR*	*b*	(*SE*)	*OR*	*b*	(*SE*)	*OR*	*b*	(*SE*)	*OR*
LGBTQ	1.073	0.148	2.925 ***	1.070	0.159	2.916 ***	1.317	0.156	3.734 ***	1.233	0.167	3.432 ***
Safe at school	--	--	--	−0.483	0.129	0.617 ***	--	--	--	−0.830	0.143	0.436 ***
Close at school	--	--	--	−0.158	0.139	0.854	--	--	--	−0.219	0.156	0.803
Community	--	--	--	−0.694	0.14	0.500 ***	--	--	--	−0.294	0.159	0.745
Depressed help	--	--	--	−0.625	0.153	0.535 ***	--	--	--	−0.527	0.173	0.591 **
Adult Help	--	--	--	−0.186	0.138	0.83	--	--	--	−0.323	0.153	0.724 *
Female	0.569	0.121	1.766 ***	0.502	0.129	1.652 ***	0.149	0.137	1.161	0.112	0.147	1.119
Grade	−0.007	0.055	0.993	0.03	0.059	1.031	−0.076	0.061	0.927	−0.055	0.065	0.946
AI/AN	−1.006	0.73	0.366	−1.237	0.879	0.29	0.978	0.699	2.658	0.338	0.848	1.401
Asian	−0.234	0.283	0.791	−0.271	0.298	0.763	−0.591	0.368	0.554	−0.552	0.385	0.576
Black/A-A	−0.299	0.138	0.741 *	−0.31	0.146	0.734 *	0.133	0.152	1.142	0.142	0.161	1.153
NH/Other PI	0.018	0.965	1.018	−0.135	0.962	0.873	2.06	1.183	7.848	2.33	1.176	10.275 *
Hisp/Latino	−0.042	0.192	0.959	−0.255	0.203	0.775	−0.03	0.231	0.971	−0.209	0.241	0.812
MR non-Hisp	0.463	0.248	1.588	0.439	0.263	1.551	0.306	0.268	0.254	0.253	0.282	1.288
Chi-square	113.502			205.424			103.17			166.106		
Log Likelihood	1653.16			1499.327			1347.065			1219.803		
Nagelkerke R^2^	0.112			0.202			0.119			0.195		
Constant	−0.859		0.423 ***	0.112		1.119	−1.055		0.348 ***	0.014		1.014

*** *p* ≤ 0.001; ** *p* ≤ 0.01; * *p* ≤ 0.05.

### 3.4. Bullied Youth

Table 4 presents the effects of bullying on depression and suicidality, as well as the effects of protective factors on these outcomes. Models 7 and 9 show that bullied youth have higher odds of reporting depression (*b* = 1.339, *OR* = 3.816, *p* ≤ 0.001) and suicidality (*b* = 0.930, *OR* = 2.535, *p* ≤ 0.001) than youth who do not experience bullying. Models 8 and 10 show that certain protective factors also matter with regard to bullying, depression and suicidality. Students who feel safe at school (*b* = −0.419, *OR* = 0.658, *p* ≤ 0.001), those who feel like they matter to their community (*b* = −0.730, *OR* = 0.482, *p* ≤ 0.001), and those who seek help most of the time or always when they feel sad, empty, hopeless, angry, or anxious (*b* = −0.590, *OR* = 0.554, *p* ≤ 0.001) have lower odds of reporting depression than students who do not. Similarly, students who feel safe at school (*b* = −0.739, *OR* = 0.478, *p* ≤ 0.001), those who seek help most of the time or always when they feel sad, empty, hopeless, angry, or anxious (*b* = −0.313, *OR* = 0.731, *p* ≤ 0.001), and those who feel comfortable seeking help from one or more adults (*b* = −0.387, *OR* = 0.679, *p* ≤ 0.05) have lower odds of reporting suicidality than students who do not. Again, these trends are similar to those presented in Table 2. When considering all of these factors, protective factors marginally moderate the effects of bullying on depression and suicidality by slightly reducing the odds ratios of each.

**Table 4 ijerph-20-06388-t004:** Protective factors, depression, and suicidality among bullied youth.

	Depression						Suicidality					
	Model 7			Model 8			Model 9			Model 10		
	*b*	(*SE*)	*OR*	*b*	(*SE*)	*OR*	*b*	(*SE*)	*OR*	*b*	(*SE*)	*OR*
Bullying	1.339	0.155	3.816 ***	1.329	0.166	3.777 ***	0.930	0.157	2.535 ***	0.846	0.167	2.331 ***
Safe at school	--	--	--	−0.419	0.131	0.658 ***	--	--	--	−0.739	0.142	0.478 ***
Close at school	--	--	--	−0.142	0.141	0.867	--	--	--	−0.245	0.156	0.782
Community	--	--	--	−0.730	0.143	0.482 ***	--	--	--	−0.313	0.159	0.731 *
Depressed help	--	--	--	−0.590	0.155	0.554 ***	--	--	--	−0.503	0.173	0.605 **
Adult help	--	--	--	−0.262	0.139	0.769	--	--	--	−0.387	0.152	0.679 *
Female	0.748	0.121	2.112 ***	0.658	0.128	1.931 ***	0.442	0.133	1.556 ***	0.378	0.143	1.460 **
Grade	0.006	0.056	1.006	0.038	0.059	1.039	−0.071	0.061	0.932	−0.058	0.065	0.944
AI/AN	−0.816	0.731	0.442	−0.819	0.867	0.441	1.013	0.707	2.755	0.699	0.808	2.011
Asian	−0.401	0.289	0.67	−0.442	0.302	0.643	−0.753	0.366	0.471 *	−0.678	0.376	0.508
Black/A-A	−0.288	0.139	0.750 *	−0.282	0.148	0.754	0.117	0.151	1.124	0.134	0.161	1.144
NH/Other PI	−0.014	0.947	0.987	−0.135	0.931	0.874	2.006	1.175	7.43	2.281	1.173	9.784
Hisp/Latino	−0.133	0.195	0.875	−0.34	0.205	0.712	−0.203	0.231	0.816	−0.349	0.24	0.706
MR non-Hisp	0.539	0.249	1.714*	0.525	0.262	1.691 *	0.431	0.261	1.539	0.38	0.277	1.463
Chi-square	139.713			225.092			67.843			134.498		
Log Likelihood	1615.537			1472.056			1354.547			1227.980		
Nagelkerke R^2^	0.137			0.221			0.081			0.162		
Constant	−1.003		0.367 ***	0.003		1.003	−1.11		0.329 ***	−0.012		0.988

*** *p* ≤ 0.001; ** *p* ≤ 0.01; * *p* ≤ 0.05.

### 3.5. Bullied LGBTQ Youth

Table 5 and Table 6 present the effects of bullying, LGBTQ identity, the interaction of bullying and LGBTQ identity, protective factors, and controls on depression and suicidality among youth. Models 11 and 14 reveal that students who are bullied (*b* = 1.259, *OR* = 3.522, *p* ≤ 0.001) and identify as LGBTQ (*b* = 0.975, *OR* = 2.651, *p* ≤ 0.001) have higher odds of reporting depression than students who are not, and students who are bullied (*b* = 0.811, *OR* = 2.249, *p* ≤ 0.001) and identify as LGBTQ (*b* = 1.203, *OR* = 3.331, *p* ≤ 0.001) have higher odds of reporting suicidality than students who are not. Models 12 and 15 highlight the interaction of LGBTQ bullying in addition to LGBTQ identity. Overall, students who are LGBTQ and experience bullying have higher odds of depression (*b* = 1.507, *OR* = 4.514, *p* ≤ 0.001) and suicidality (*b* = 0.826, *OR* = 2.284, *p* ≤ 0.001) than students who are not LGBTQ and bullied. Students who identify as LGBTQ have higher odds of depression (*b* = 0.701, *OR* = 2.016, *p* ≤ 0.001) and suicidality (*b* = 1.028, *OR* = 2.794, *p* ≤ 0.001). Models 13 and 16 show that certain protective factors lower the odds of reporting depression and suicidality when considering all covariates. Overall, students who are bullied have higher odds of depression (*b* = 1.177, *OR* = 3.243, *p* ≤ 0.001) and suicidality (*b* = 0.712, *OR* = 2.038, *p* ≤ 0.001) than students who are not bullied, and students who identify as LGBTQ have higher odds of reporting depression (*b* = 0.908, *OR* = 2.479, *p* ≤ 0.001) and suicidality (*b* = 1.103, *OR* = 3.012, *p* ≤ 0.001) than students who are not LGBTQ. In terms of protective factors, students who feel safe at school (*b* = −0.389, *OR* = 0.678, *p* ≤ 0.05), those who feel like they matter to their community (*b* = −0.707, *OR* = 0.493, *p* ≤ 0.001), and those who seek help most of the time or always when they feel sad, empty, hopeless, angry, or anxious (*b* = −0.612, *OR* = 0.542, *p* ≤ 0.001) are less likely to report depression than students who do not. In turn, students who feel safe at school (*b* = −0.736, *OR* = 0.479, *p* ≤ 0.001), those who seek help most of the time or always when they feel sad, empty, hopeless, angry, or anxious (*b* = −0.514, *OR* = 0.598, *p* ≤ 0.01), and those who feel comfortable seeking help from one or more adults (*b* = −0.379, *OR* = 0.685, *p* ≤ 0.05) are less likely to report suicidality than students who do not.

**Table 5 ijerph-20-06388-t005:** Protective factors and depression among bullied LGBTQ youth.

	Depression								
	Model 11			Model 12			Model 13		
	*b*	(*SE*)	*OR*	*b*	(*SE*)	*OR*	*b*	(*SE*)	*OR*
**Bullying**	1.259	0.158	3.522 ***	--	--	--	1.177	0.191	3.243 ***
LGBTQ × Bullying	--	--	--	1.507	0.341	4.514 ***	0.371	0.420	1.449
LGBTQ	0.975	0.152	2.651 ***	0.701	0.165	2.016 ***	0.908	0.181	2.479 ***
Safe at school	--	--	--	--	--	--	−0.389	0.133	0.678 **
Close at school	--	--	--	--	--	--	−0.126	0.144	0.882
Community	--	--	--	--	--	--	−0.707	0.145	0.493 ***
Depressed help	--	--	--	--	--	--	−0.612	0.157	0.542 ***
Adult Help	--	--	--	--	--	--	−0.254	0.142	0.776
Female	0.580	0.125	1.787 ***	0.584	0.122	1.792 ***	0.161	0.15	1.174
Grade	0.009	0.057	1.009	0	0.055	1.000	−0.052	0.066	0.949
AI/AN	−1.134	0.764	0.322	−1.172	0.778	0.310	0.453	0.861	1.574
Asian	−0.239	0.292	0.788	−0.234	0.284	0.791	−0.492	0.383	0.611
Black/A-A	−0.258	0.142	0.773	−0.316	0.139	0.729 *	0.169	0.165	1.185
NH/Other PI	−0.006	0.963	0.994	0.077	0.949	1.080	2.296	1.175	9.936
Hisp/Latino	−0.014	0.198	0.986	−0.052	0.193	0.949	−0.21	0.246	0.811
MR non-Hisp	0.474	0.256	1.606	0.424	0.251	1.528	0.276	0.285	1.317
Chi-square	181.314			136.167			262.575		
Log Likelihood	1573.936			1628.610			1434.573		
Nagelkerke R^2^	0.175			0.133			0.254		
Constant	−1.145		0.318 ***	−0.874		0.417 ***	−0.161		0.851

*** *p* ≤ 0.001; ** *p* ≤ 0.01; * *p* ≤ 0.05.

**Table 6 ijerph-20-06388-t006:** Protective factors and suicidality among bullied LGBTQ youth.

	Suicidality								
	Model 14			Model 15			Model 16		
	*b*	(*SE*)	*OR*	*b*	(*SE*)	*OR*	*b*	(*SE*)	*OR*
**Bullying**	0.811	0.163	2.249 ***	--	--	--	0.712	0.207	2.038 ***
LGBTQ × Bullying	--	--	--	0.826	0.299	2.284 **	0.082	0.381	1.085
LGBTQ	1.203	0.161	3.331 ***	1.028	0.179	2.794 ***	1.103	0.197	3.012 ***
Safe at school	--	--	--	--	--	--	−0.736	0.146	0.479 ***
Close at school	--	--	--	--	--	--	−0.218	0.16	0.804
Community	--	--	--	--	--	--	−0.278	0.162	0.758
Depressed help	--	--	--	--	--	--	−0.514	0.176	0.598 **
Adult Help	--	--	--	--	--	--	−0.379	0.156	0.685 *
Female	0.199	0.141	1.220	0.179	0.138	1.196	0.161	0.15	1.174
Grade	−0.068	0.063	0.934	−0.069	0.062	0.933	−0.052	0.066	0.949
AI/AN	0.835	0.747	2.304	0.806	0.738	2.239	0.453	0.861	1.574
Asian	−0.534	0.369	0.586	−0.568	0.366	0.567	−0.492	0.383	0.611
Black/A-A	0.164	0.155	1.179	0.118	0.153	1.126	0.169	0.165	1.185
NH/Other PI	2.063	1.191	7.870	2.09	1.177	8.081	2.296	1.175	9.936
Hisp/Latino	−0.046	0.237	0.955	−0.043	0.232	0.958	−0.21	0.246	0.811
MR non-Hisp	0.331	0.272	1.393	0.291	0.27	1.338	0.276	0.285	1.317
Chi-square	124.241			104.133			178.055		
Log Likelihood	1298.148			1332.247			1184.424		
Nagelkerke R^2^	0.144			0.121			0.21		
Constant	−1.286		0.276 ***	−1.075		0.341 ***	−0.209		0.812

*** *p* ≤ 0.001; ** *p* ≤ 0.01; * *p* ≤ 0.05.

## 4. Discussion

This study found that individual, school, and community-level factors moderate depression and suicidality among youth. Overall, about 22% of youth in this sample identified as LGBTQ, which is higher than the national reported averages [1] and affirms the need for the recognition that queer youth are an ever-growing population whose needs must be accounted for [5,6]. Poor mental health is a serious issue among youth, with 38% of the sample reporting depression and 28% of the sample reporting suicidality; these are numbers that are well above the national averages according to the National Institute of Mental Health [27]. One of the most notable findings is that, despite higher-than-average percentages of youth reporting depression and suicidality, Table 1 shows that only 26% of youth in the sample sought help when they felt depressed. Studies show that youth may not seek help even when they are struggling with poor mental health, because of negative attitudes towards mental illness and the stigma surrounding help-seeking behaviors [28]. Reducing shame and stigma surrounding help-seeking behaviors must be a point of concern when fostering positive mental health and wellbeing among young people. Schools must take the lead on promoting mental health awareness and campaign to normalize help-seeking behaviors individually and institutionally.

There are consistencies in protective factors for both depression and suicidality, which warrant recognition. When students feel safe at school, feel that they matter to their community, and seek help when they feel depressed and from an adult, it significantly reduces the odds of reporting depression or suicidality as a result of bullying and LGBTQ identity. This coincides with the literature and affirms findings from other major surveys [20,21]. More importantly, however, is the fact that this illuminates the significance of a multilevel approach to supporting youth; this is an approach that considers personal, familial, school, and community-level factors. In addition to promoting positive attitudes about mental health, school systems must also work to improve their climates. The 2021 National School Climate Survey recommends that in order to improve the school climate of LGBTQ youth, schools must include LGBTQ history and curriculum, supportive clubs (such as GSAs), provide professional development for school staff and administration, support gender-affirming school policies and practices, and provide comprehensive bullying and harassment policies that protect LGBTQ students [8]. All of these factors would affirm safer school environments, and families, schools, and communities must move towards implementing these aspects in order to create safer environments for students, especially LGBTQ youth.

Borrowing from the *Whole School, Whole Child, Whole Community* (WSCC) model created by the CDC [29]—an ecological approach to learning and health that links a child’s needs to both school and the community—these findings highlight the importance of ensuring that youth feel safe and supported in their communities, as well as the school system. The spectrum of community support can range from simply hanging a pride flag or a ‘Black Lives Matter’ sign outside one’s house in order to signal visibility and a safe community for young people, to local political action. Creating community support systems, safe spaces, and activism are a huge building block for helping youth feel not only accepted, but also truly wanted and supported in their communities. In creating these spaces, schools and communities could naturally ensure that youth, especially LGBTQ youth, have adequate mentorship and guidance to be able to talk about the issues they may face. All of these things would, in turn, help improve the lives of young people.

### Limitations

While this study contributes to the broader literature, it does not exist without certain limitations. First, the YRBSS secondary data are limited in the scope of questions that are asked across districts and states. For example, the protective factor questions examined in this study are considered “optional” questions and are thus, not required to be included in surveys. Future YRBSS questionnaires should require that these questions be included. Second, the YRBSS survey does not collect school-level identifiers, and future research could benefit greatly from nested models that examine these issues within geographically identified regions with regard to the questions at hand. Third, the results showed that gender, race, and ethnicity matter with regard to depression and suicidality among youth; thus, future studies should push to be more intersectional in their understanding of the issue at hand. Despite these limitations, this study confirms the importance of acknowledging and cultivating protective factors that can benefit youth.

## 5. Conclusions

Lesbian, gay, bisexual, transgender, queer and/or questioning (LGBTQ) youth are a growing population in the United States, and society must acknowledge the issues that they face, rather than work to perpetuate them. In this study, whether or not students were bullied or identified as LGBTQ, the odds ratios for depression and suicidality reduced when students felt safe at school, felt that they mattered to their community, sought help when they felt depressed, and could seek help from an adult other than their parents. Thus, schools and communities must move towards implementing strategies of support in the face of current state-level legislation that attempts to nullify their importance and visibility. While the purpose of this study was not to measure policy, it can highlight the need to halt the trajectory of the record number of state-level anti-LGBTQ legislations that are currently being put forth. The institutionalization of denying youth the ability to exist wholly as themselves desecrates feelings of protection and belonging, and as a result, increases the possibility of risk and harm to LGBTQ youth. Our goal as a society should be to continue building protection, connection, and belonging for youth, especially those most vulnerable.

## Data Availability

https://www.cdc.gov/healthyyouth/data/yrbs/data.htm (accessed on 5 February 2023).

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
