# Peer review of "Protection and Connection: Negating Depression and Suicidality among Bullied, LGBTQ Youth"

_ijerph, 2023, doi:10.3390/ijerph20146388_

Round 1
Reviewer 1 Report
It is an interesting study, and the manuscript has potential. Nonetheless, the manuscript entails important unclarities and some sections need a clearer structure. Please find details below.
The abstract should be more specific about the methods of analysis and the results.
The introduction is interesting but includes many statements that are not supported by literature. Also, statements that say ‘the literature’ or this ‘survey’ found that … should be supported by references. Please check throughout the manuscript.
The introduction refers to the ‘2015-2019 Youth Risk Behavior Survey’. Is this the same survey that was used for analysis? If so, please be consistent in the name used for the survey.
These sentences at the end of the introduction can be deleted as this is part of the results, and should not be included in the introduction.
“Overall, results show that feeling safe at school, feeling like youth matter to their communities, and having positive help seeking behaviors, do in fact moderate the likelihood of depression and suicidality among youth. The implications of these findings are discussed.”
Materials and Methods
Please specify the age range of the sample.
Data analysis
This section is very unclear. At the end of the introduction, the authors specified four research questions. Please organize the ‘data analysis’ according to these four research questions.
Also, be more clear about what type of analyses (e.g., which type of regression analysis) will be carried out, and confirm whether or not the conditions for these type of analyses had been met.
Results
Tables should be reported in the Results section.
A table should be included in the manuscript after the paragraph in which the table has been mentioned the first time.
The Results section should be organized according to the four research questions, so that the reader can see what the findings are according to the aims of the study. Currently, this is not clear.
Discussion
The Discussion should also follow the same structure.
The discussion section could be more in-depth by discussing the study findings in the context of other literature. Currently, it reads more like a summary of the findings with a few comments. I would encourage the authors to dig deeper into the literature to enrich the discussion and to make it more relevant for the reader.
There seems to be something wrong with the referencing system as some of the numbers in the reference list include multiple references.
English is reasonable. Maybe check for typos?
Author Response
Please see attached for reviewer responses.

Reviewer 2 Report
Dear authors,
please see the attached file with my comments

Author Response

(The authors gave the same response as above.)

Round 2
Reviewer 2 Report
Dear author(s),
I had the opportunity to read your revised version of the above manuscript. I realized that you have addressed in proper way almost all the issues I raised in my initial review
Please see below some minor additional comments on specific points of the revised manuscript.
1. Introduction
lines 30-34: “Stigma …… in (not just physical) violence”: Authors must provide references for these two sentences: on stigma and institutional discrimination.
Personal experience is valuable but statements like the ones above must be evidence-based. There is plenty of literature regarding institutional discrimination of LGBTQ people and its consequences to people’s lives. Please provide some references.
Lines 84-85: The sentence remained the same (“One of the biggest proponents of the LGBTQ protective factors literature deals with promoting resilience, or the capability that people have to overcome adversity.”). It still doesn’t make sense. Please rephrase
2. Materials and Methods
Line 172: Change to “measured”. My comment was related to the usage past tense! Sorry for the confusion regarding the line
Authors, please remove all Table at the Results section, so that they are as close as possible to the paragraphs which verbally present the finding. For example, Table 1 must be at the sub-section 3.1 Descriptive Statistics
Author Response
Please see attached document for comments.
